# Self-reported nursing competence among registered nurses in Jordan: A cross-sectional study

Alaadin Alharaizahe[1], Intima Alrimawi[2]*, Hekmat Yousef Al-Akash[1],
Abedalmajeed Shajrawi[3], Abdul-Monim Batiha[4], Osama Al-Kouri[5], Manar Abu-Abbas[5],
Haitham Khatatbeh[5], Ahmad Rajeh Saifan[5]

1 Clinical Nursing Department, Faculty of Nursing, Applied Science Private University (ASU), Amman,
Jordan, 2 Berkley School of Nursing, Georgetown University Medical Center, Georgetown University,
Washington DC, United States of America, 3 Faculty of Health Science, Higher College of Technology,
Abu Dhabi, United Arab Emirates, 4 College of Nursing -Mahyel Asir, King Khalid University, Saudi Arabia,
5 College of Nursing, Yarmouk University, Irbid, Jordan

* ia409@georgetown.edu

journal.pone.0341714

University, JORDAN

**Peer Review History:** PLOS recognizes the
benefits of transparency in the peer review
process; therefore, we enable the publication
of all of the content of peer review and
author responses alongside final, published
articles. The editorial history of this article is
available here: https://doi.org/10.1371/journal.
pone.0341714

## Abstract

### Background

Nurses constitute a significant portion of Jordan's healthcare workforce, and their
competence plays a critical role in patient safety and quality of care. Global and
national strategies highlight the importance of assessing and improving nursing com-
petence. Despite the critical role of nurses, research on self-reported nursing compe-
tence (SRNC) in Jordan remains limited.

### Aim

This study explores self-reported nursing competence (SRNC) among registered
nurses (RNs) in Jordan.

### Methods

A cross-sectional study was conducted with a sample of 212 RNs from public, pri-
vate, and teaching hospitals.. Data were collected using the short form of the Nursing
Professional Competence (NPC) scale, a 35-item instrument scored on a 7-point
Likert scale ranging from 1 (very low competence) to 7 (very high competence), with
higher scores indicating greater perceived professional competence. The derived
data was analyzed using SPSS software.

### Results

The overall level of self-reported nursing competence was high, with a mean score
of 76.69%. Among these six domains, the domain of "documentation and adminis-
tration of care" scored the highest (78.26%), and "care pedagogy" scored the lowest

**Data availability statement:** All relevant data are within the paper and its Supporting Information files.

**Funding:** Funding support is in part from Georgetown University Medical Center.

**Competing interests:** The authors have declared that no competing interests exist.

(75.23%). The items that scored high included "respectful communication," "clinical follow-up," and "documentation," which scored "to a high degree." The items that scored lower included "psychological needs," "group-based education," and "respect for different values," which scored "to a fairly high degree.".".

## Conclusions

The Jordanian nursing community views itself as very competent; however, improvements are needed in particular fields. Continuous learning and training are required in order to retain and upgrade professional competency. The findings highlight the value of mentorship, a supportive work environment, and continuing education. Future research should include peer and supervisor evaluations to validate self-reported competence and better guide workforce development strategies.

## Introduction

The professional competence of Registered Nurses (RNs) is an important issue in health care as it relates to professional standards, patient safety and the quality of nursing [1]. It indicates that nurses are performing their care based on specific preset standards issued by credentialing regulated bodies in the field of health and nursing [2]. This was guided by many recent changes in health care that led to increased demand on RNs with respect to their quality or professional competence as well as their quantity.

Nursing Professional Competence (NPC) involves the integration of skills, knowledge, attitudes, and abilities that can ensure a high level of professional performance [3]. A positive relationship between the high NPC level of RNs and the patient safety, health status, low morbidity and mortality levels as well as adverse events is well established in literature [4]. Additionally, higher competence level was positively associated with higher levels of job satisfaction [5], while less competent RNs were reported to be more frustrated and dissatisfied that can impact their patients care [6,7]. Low levels of NPC among RNs have been shown to increase the morbidity and mortality of inpatients [8].

Jordan has a pluralistic healthcare system composed of public, private, and teaching sectors. The public sector, with the Ministry of Health (MOH) and the Royal Medical Services (RMS), still provides the greatest number of healthcare services and caters to the largest number of people in the country [9]. Although there has been significant progress in the health status of the people and in the progress towards universal health coverage, the healthcare system in Jordan remains faced with ongoing challenges such as shortages of healthcare workers, rising demand for healthcare services due to the rising number of people and the influx of refugees, financial and resource challenges, and regional inequalities in access to care and the quality of care provided [9,10]. In this respect, recent health plans in the country, in conformity with the guidelines of the World Health Organization (WHO), focus on improving the quality of care, capacity building, and strengthening the health system, with the importance of nursing competency in ensuring the provision of safe, effective, and equitable healthcare [9]

No study concerning the views of Jordanian nurses about their own professional competence could be located. Accordingly, this study was planned to be conducted in order to fill this gap in literature. The findings of this study can serve as baseline data for future research in this field. Enhancement of nurses' competence can improve patient care, increase patient satisfaction, decrease morbidities and complications, and decrease patients' length of hospitalization [4]. In Jordan, nursing education programs are traditional, not responsive to the national health needs and challenges, and lack innovative approaches in education, such as competency-based education, technology and information, and partnership with service institutions [11]. The quality of nursing and midwifery graduates does not meet the population health needs and advances in the health care system [12]. The assessment of the nurse's clinical competence can help in determining the educational needs of nurses, identifying the fields that require further improvement, and consequently, a satisfactory delivery of nursing care. Moreover, the curricula and hospital policies based on various accreditation standards require timely revision, update, or modification. This dynamic change in programs' curricula and hospitals' standards of care may influence the SRPNC levels among RNs that have to be periodically evaluated.

## The purposes of the study

Assess the self-reported nursing competence (SRPNC) among registered nurses in Jordan.

## Research question

1. What is the self-reported level of professional nursing competence among registered nurses in Jordan across different domains of practice, as measured by the Nursing Professional Competence (NPC) scale?

## Literature review

**Definitions of nursing competence (NPC).** The literature on the concept of competence reveals three main approaches to its conceptualization: (1) the behavioristic approach, which views competence as a set of tasks and skills; (2) the generic approach, which emphasizes transferable attributes; and (3) the holistic approach, which integrates knowledge, skills, attitudes, and values [7,13,14]. In nursing, competence is defined as the professional standards that guide clinical practice [15], as well as the ability to integrate knowledge, skills, attitudes, and values [16]. According to these definitions, nursing competencies have three major foundations: knowledge, clinical skill, and moral areas including; attitude and values [14]. Recent evidence further demonstrates that nursing competence is closely associated with emotional intelligence and critical thinking abilities, which are essential for effective clinical judgment, professional communication, and patient-centered care, highlighting the importance of educational approaches that foster these competencies [17,18]Furthermore, competence includes professional actions, adherence to standards, and the embodiment of the professional role [18].

Moreover, regulatory bodies have sought to define the concept of competence to achieve international consensus. For example, the Australian Nursing and Midwifery Council describes "competence" as the combination of skills and knowledge; what you believe; your ability to understand what something means in the context of your job; and your ability to use judgment in making decisions about your work [19,20]. They expect that all nursing graduates will be able to demonstrate a level of competence that is sufficient to provide safe and effective nursing care to their patients. In addition, nursing graduates must possess enough competent registered nurses to provide a quality of care that is safe, high-quality, and patient-centered care [4].

## Nursing professional competence (NPC) assessment

Numerous studies have been conducted to assess NPC among nurses upon graduation [21–27]. A study was conducted in Sweden by Gardulf et al. [21] found that nurses at the point of their graduation reported high competence levels in patient-related nursing, medical and technical care, documentation and information technology, and value-based nursing.

Female nurses reported higher levels of professional competence in these areas. Additionally, a cross-sectional study in Iran assessed the clinical competency of undergraduate nursing students using the Self-assessment Clinical Competence Questionnaire. Five factors were extracted from the questionnaire, including technical competence, advanced competence, ethical competence, care management and safety competence [28].

Moreover, several studies assessed the SRC of nursing at the point of their graduation and novice stage of their professional life (newly graduated nurses) using NCS and NPC scales [21,29,30]. It was found that SRC was rated as quite good on the NCS scale (NCS scores between 25–50 based on visual analogue scale 1–100) [30], good in others (NCS scores between 50–75) [31,32], and high in one of the studies on NPC (NPC scale scores between 69–90) [21].

These variable findings are attributed to many factors such as variable assessment tools, high scores on NPC might be explained by that graduates are expected to overestimate their competence due to lack of exposure to clinical practice during their studies [27], while newly graduated nurses are thought to be anxious and uncertain about the level of their professional competence, non-coincident and worry about making mistakes, lacking sufficient management and technical skills, and insufficiently prepared due to the incongruence between educational standards and the real health care requirements [33].

### Self-assessment of nurse competence

Self-assessment is the most common method to assess nurse competence in the clinical context and is considered a time-saving and cost-effective method [34,35]. Self-assessment scales can help people develop meta-cognitive skills, assist them in becoming self-learners and self-confident. Self-assessment can also empower staff and help them to pick higher professional goals and to try harder to realize these goals [36]. SRNPC has been used in many studies to investigate nurses' current levels of competence and the factors related to it [21,37,38]. Self-assessment of nursing competence is important, as it forms a critical component of continuous professional development [39], and it impacts on their job behaviors such as turnover intention and burnout [40].

The literature review highlights that assessing nurses' competence is a significant concern within the nursing profession. Numerous studies have been conducted in countries to examine nurses' levels of competence, with most relying on self-reported instruments. However, studies exploring nurses' perceptions of their own professional competence in Jordan are lacking. This study was therefore conducted to assess the competence levels of Jordanian nurses.

## Methodology

### Design

A cross-sectional correlational design was used for the current study. In quantitative research, the choice of design depends on the experience of the researcher, the problem statement, and the purpose of the research [41]. This approach was chosen because it is the best design for this study, relatively quick, cheap and easy to conduct (no long periods of follow-up). Data on all variables is only collected once.

### Setting

The healthcare system in Jordan is divided into four major sectors: the governmental sector, affiliated with the Ministry of Health, the Royal Medical Services (RMS), the teaching sector, and the private sector [42]. For the purpose of this study, five hospitals were purposively selected to reflect the diversity of the healthcare system and address the research questions. These included one governmental hospital, three private hospitals, and one teaching hospital. Al-Basheer Hospital is the largest government hospital in Jordan and has a total of 1,100 beds. Al-Basheer Hospital is located in Amman. In the list of selected private hospitals, all located in Amman, are Istishari Hospital, with a total of 140 beds, Istiklal Hospital, with a total of 200 beds; and Ibn Alhaytham Hospital, with a total of 200 beds. In the list of selected teaching hospitals for

this research is King Abdullah University Hospital. King Abdullah University Hospital owns a total of 683 hospital beds and is the educational hospital for the Faculty of Medicine of Jordan University of Science and Technology.

## Population

The study's target group was registered nurses in Jordan, working in governmental, private, and teaching hospitals in Jordan. Criteria for participation in this study included being of Jordanian nationality, possessing a Bachelor of Science in Nursing degree, and being fully employed in one of the following hospitals: governmental, private, or teaching. Participants were not to be nurses in an administrative position, such as managers, head nurses, or supervisors. The study focused exclusively on bedside RNs who provide direct patient care, excluding those in managerial roles without regular patient contact.

## Sample

Convenient sampling technique was used to select the study sample of nurses' work at the three sectors (public, private, and teaching). The researcher used this method to be able to pull a large number of readily available nurses on duty in a fast, easy, and cost-effective manner [43]. To make sure that the sample size is sufficient to get statistical significance, sample size is determined using G* power 3.1 software [44]. A power of 0.90, an alpha level of 0.05, and a medium effect size of 0.03 were used in calculating the sample size. Based on these assumptions, the estimated sample size is 192 nurses. Drop out of participants, non-response, or no return of the questionnaire are expected in the research. The researcher added 10% (19 subjects) for the estimated sample size. Accordingly, the final sample size was 212 nurses.

## Instrumentation

To enhance response rates and reduce completion time, this study used the validated short version of the NPC scale (35 items), developed by Nilsson et al. [24].. The NPC scale assesses six key areas of nursing competence: nursing care, value-based nursing care, medical and technical care, care pedagogics, documentation and administration of nursing care, and development, leadership and organization of nursing care. Reliability for the short version ranged from 0.71 to 0.86. The NPC scale used a seven-point Likert scale to measure competence, ranging from 1 (very low competence) to 7 (very high competence). Analyzed mean scores were calculated to obtain percentages, marking very low competence <30 percent, low competence at 30–49 percent, moderate competence at 50–68 percent, high competence 69–90 percent, and very high competence >90 percent competence level [21]. The data-gathering process also comprised a demographic survey to get data on respondents' characteristics, including their age, gender, experience, education level, exposure to training, and characteristics of their hospital.

NPC-SR (Self-Reported Nurse Professional Competence Scale) started out as work by Nilsson et al. [26] to assess self-reported professional competence among nursing students and registered nurses (RNs). It consists of 88 items across eight competence areas: Nursing Care, Value-Based Nursing, Medical and Technical Care, Teaching/Learning and Support, Documentation and IT, Legislation and Safety, Leadership and Education Development, and Supervision. The scale has shown strong reliability, with Cronbach's alpha values ranging from 0.75 to 0.94 for individual areas and 0.97 for the full scale used internationally in studies examining competence at graduation, after curriculum changes, or in relation to professional experiences such as conflict or disaster management [21,23,25].

## Pilot testing

To ensure reliability and clarity of the questionnaire, pilot testing was applied to 10% (22 nurses) of the total sample of similar characteristics to the original sample within the same settings. The subjects included in the pilot testing were excluded in the final analysis. Reliability of the questionnaire was evaluated and gave a result of 0.87 reliability coefficient.

## Reliability

It was found that the reliability coefficient (Cronbach's alpha) in this study ranges from 0.764 (which is an acceptable value) to 0.927 (which is a high value). In general, the reliability coefficient for all items is 0.963 which is considered to be a very high value showing a reliable measure of the study tool.

## Ethical consideration

Ethical approval was obtained from the research and ethical committee at the Applied Science Private University (ASU), and Institutional Review Board (IRB) approval was obtained before data collection from all sites. All participants' rights were ensured based on ethical principles of respect for human dignity, privacy, confidentiality, and autonomy. Participation in the study was voluntary. All participants were informed about the purpose of the study, study procedures, potential risks and benefits, and their right to decline participation or withdraw from the study at any time without any consequences. Written informed consent was obtained from all participants prior to completing the questionnaire. After data collection, to ensure confidentiality, all data were coded and entered into a password protected computer. Only the principal investigator and the research team have access to it.

## Data collection procedure

The principal investigator collected the data from August 1 to December 31, 2019, after obtaining the IRB approval and meeting with all the nursing directors. Then all nursing directors were asked to communicate with the head nurses to inform them that the researcher will collect the data. Nurses were asked to participate voluntarily and anonymously by signing the consent form after a thorough explanation of the study by the principal investigator. After obtaining the consent, final arrangements were made with the supervisors to arrange a proper time to carry out the study. Questionnaires were distributed to the nurses over the three shifts by the principal investigator at the beginning of each shift. All questions from the participants regarding the data collection instruments were answered by the principal investigator. After the participants have completed the questionnaires, the principal investigator has collected them at the end of each shift.

## Data analysis

Data was entered and analyzed using SPSS version 24. Means, standard deviations, frequencies, and percentages were used to describe demographic and professional data of participants and evaluate nursing professional competence (NPC) in six domains. Data analysis was conducted per domain separately, with mean values and standard deviations being computed at items per domain, whereas overall percentage scores were calculated to denote average competence per domain. Interpretation of mean scores was made according to guidelines from the scale developer and followed "to a high degree" or "to a fairly high degree" categorization. Following recommendations outlined by Nilsson et al. [24], questionnaires with less than 60% completion of NPC items were excluded from the analysis and considered as dropout.

## Result

### Sample description

A total of 212 nurses participated in the study, with a 100% response rate. Of the sample, 120 nurses (56.6%) were female, with a mean age of 30.03 years (SD=4.92). Participants demonstrated a wide range of nursing experience, with a mean of 7.40 years (SD=4.77). A third of the sample had minimal nursing experience (0–4 years; n=70, 33.0%), while over one-fourth (31.1%) had 5–8 years of experience (n=66). Approximately one-fifth of nursing participants (n=44, 20.8%) had 9–12 years of experience in nursing, and only 15.1% (n=32) reported having more than 12 years of experience. The sample included both less experienced and more experienced nurses based on these numbers. Out of the 212 total participants included in the analysis, most (n=154, 72.6%) had experience with both orientation and mentoring programs prior to entering practice. Over half of the nurses (n=124,

59.6%) graduated with a "good" academic grade, and 85 nurses (40.3%) worked in medical or surgical departments. A total of 88 nurses (41.5%) were employed in public hospitals, and nearly two-thirds (76.2%) were graduates of public universities. Additionally, more than half of the nurses (n = 122, 57.5%) worked in accredited hospitals, see Table 1.

### Levels of NPC

Table 2 presents the findings of the nursing care domain of NPC. It is evident that both item (4), "Document the patient's physical condition," and item (1), "Independently apply the following stages in the nursing process: observation and assessment (nursing anamnesis, status, and nursing goals)," are ranked as the highest areas for NPC because they have

**Table 1. Demographic and clinical setting characteristics of the study sample.**

| Variable | Category | N | % |
|---|---|---|---|
| Age (30.03±4.92) | 22-26 years | 54 | 25.50 |
| | 27-31 years | 86 | 40.60 |
| | 32-36 years | 44 | 20.80 |
| | more than 36 years | 28 | 13.20 |
| Gender | Male | 92 | 43.40 |
| | Female | 120 | 56.60 |
| Experience (7.40±4.77) | 0-4 years | 70 | 33.00 |
| | 5-8 years | 66 | 31.10 |
| | 9-12 years | 44 | 20.80 |
| | more than 12 years | 32 | 15.10 |
| Exposure to orientation program | No | 38 | 17.90 |
| | Yes | 174 | 82.10 |
| Exposure to mentoring program | No | 58 | 27.40 |
| | Yes | 154 | 72.60 |
| Degree grades (n = 208) | Satisfactory | 27 | 13.00 |
| | Good | 124 | 59.60 |
| | Very Good | 49 | 23.60 |
| | Excellent | 8 | 3.80 |
| Clinical area (n = 211) | Medical, Surgical | 85 | 40.30 |
| | OPD, Day Case | 2 | 0.90 |
| | ICU,CCU,NICU, PICU | 40 | 19.00 |
| | ER | 23 | 10.90 |
| | Pediatric | 29 | 13.70 |
| | Oncology | 18 | 8.50 |
| | OR | 14 | 6.60 |
| Hospital's sector | Public | 88 | 41.50 |
| | Private | 54 | 25.50 |
| | Teaching | 70 | 33.00 |
| University (n = 210) | Private | 50 | 23.80 |
| | Public | 160 | 76.20 |
| Undergraduate Curricula | Sufficient | 194 | 91.50 |
| | Insufficient | 18 | 8.50 |
| Hospital's accreditation | Accredited | 122 | 57.50 |
| | Not Accredited | 90 | 42.50 |

**Table 2. Mean, SDs, percentage, & interpretation of the nursing care subscale.**

| Rank | Item No. | Item | Mean | SD | Attitude |
|---|---|---|---|---|---|
| 1 | 4 | Document the patient's physical condition? | 5.44 | 1.38 | To a fairly high degree |
| 2 | 1 | Independently apply the following stages in the nursing process: observation and assessment (nursing anamnesis, status and nursing goals)? | 5.44 | 1.42 | To a fairly high degree |
| 3 | 2 | Cater for the patient's needs regarding basic, physical nursing care? | 5.43 | 1.32 | To a fairly high degree |
| 4 | 3 | Cater for the patient's needs regarding specific, physical nursing care? | 5.26 | 1.37 | To a fairly high degree |
| 5 | 5 | Document the patient's psychological condition? | 5.03 | 1.39 | To a fairly high degree |
| **Nursing care %** | | | **76.03** | **16.53** | **High** |

the highest mean of 5.44 out of 7.0, indicating a nurse attitude of "to a fairly high degree." In contrast, item (5), "Document the patient's psychological condition," has the lowest mean at only 5.03 ± 1.39, reflecting an attitude of "to a fairly high degree." The overall percentage score for the nursing care domain was 76.03%, corresponding to a high level of competence.

From Table 3 the analysis of the value-based nursing care area showed that item (6), "Communicate with patients, next of kin, and staff respectfully, sensitively, and empathetically?" has the highest mean of 5.44 ± 1.42 out of 7.0 with an attitude of "to a fairly high degree." In the second rank, item (10), Utilize the knowledge and experience of the team and others, and through team collaboration contribute to a holistic view of the patient," has a mean of 5.42 ± 1.33 and an attitude of "to a fairly high degree." Item (9), "Show openness to and respect for different values and faiths?" has the lowest mean of 5.27 ± 1.45 with a "to a fairly high degree" attitude. The overall percentage of value-based nursing care is 76.56%, classified as high.

With regard to the medical and technical care area, Table 4 demonstrates that the item (15), "Follow up on the patient's condition after examinations and treatments?" has the highest mean of 5.42 ± 1.37 with the attitude of "to a fairly high degree." In the second rank, item (14), "Display judgment, knowledge, and thoroughness when informing and providing

**Table 3. Mean, SDs, percentage, & interpretation of the value-based nursing care subscale.**

| Rank | Item No. | Item | Mean | SD | Attitude |
|---|---|---|---|---|---|
| 1 | 6 | Communicate with patients, next of kin and staff respectfully, sensitively and empathetically? | 5.44 | 1.42 | To a fairly high degree |
| 2 | 10 | Utilize the knowledge and experience of the team and others, and through team collaboration contribute to a holistic view of the patient? | 5.42 | 1.33 | To a fairly high degree |
| 3 | 7 | Show concern and respect for the patient's autonomy, integrity and dignity? | 5.37 | 1.42 | To a fairly high degree |
| 4 | 8 | Utilize the knowledge and experience of the patient and/or their next of kin? | 5.30 | 1.30 | To a fairly high degree |
| 5 | 9 | Show openness to and respect for different values and faiths? | 5.27 | 1.45 | To a fairly high degree |
| **Value-Based Nursing Care %** | | | **76.56** | **16.60** | **High** |

**Table 4. Means, SDs, percentages, & interpretations of the medical and technical care area.**

| Rank | Item No. | Item | Mean | SD | Attitude |
|---|---|---|---|---|---|
| 1 | 15 | Follow up the patient's condition after examinations and treatments? | 5.42 | 1.37 | To a fairly high degree |
| 2 | 14 | Display judgment, knowledge and thoroughness when informing and providing for the patient's security and wellbeing during examinations and treatments? | 5.37 | 1.37 | To a fairly high degree |
| 3 | 11 | Manage drugs adequately, applying knowledge in pharmacology? | 5.34 | 1.39 | To a fairly high degree |
| 4 | 16 | Handle medical products on the basis of existing regulations and safety routines? | 5.33 | 1.44 | To a fairly high degree |
| 5 | 12 | Independently administer prescriptions? | 5.31 | 1.44 | To a fairly high degree |
| 6 | 13 | Question unclear instructions/prescriptions? | 5.17 | 1.52 | To a fairly high degree |
| **Medical and Technical Care %** | | | **76.03** | **16.20** | **High** |

for the patient's security and wellbeing during examinations and treatments?" has a mean of $5.37 \pm 1.37$ and an attitude of "to a fairly high degree." Item (13), "Question unclear instructions/prescriptions?" has the lowest mean of $5.17 \pm 1.52$ with a "to a fairly high degree" attitude. The overall percentage of medical and technical care is 76.03%, corresponding to a high level of competence.

For care pedagogics, item (21), "In dialogue, motivate the patient to comply with treatments?" has the highest mean, with $5.39 \pm 1.29$ and an attitude of "to a fairly high degree." In the second rank, item (18), "Inform and educate patients and next of kin individually, taking into account time, form, and content," has a mean of $5.30 \pm 1.61$ and an attitude of "to a fairly high degree." Item (19), "kin in a group, taking into account time, form, and content?" has the lowest mean of $5.13 \pm 1.65$ with an attitude of "to a fairly high degree." The overall percentage of care pedagogics is 75.23%, categorized as high; see Table 5.

From Table 6, the documentation and administration of nursing care, item (26), "Handle sensitive information correctly and carefully?" has the highest mean of $5.68 \pm 1.30$ with an attitude of "to a fairly high degree." In the second rank, item (28), "Continuously engage in your own personal and professional competence development?" has a mean of $5.55 \pm 1.26$ and an attitude of "to a fairly high degree." Item (23), "Use information and communication technology to support nursing

**Table 5. Mean, SDs, percentage, & interpretation of the care pedagogics area.**

| Rank | Item No. | Item | Mean | SD | Attitude |
|---|---|---|---|---|---|
| 1 | 21 | In dialogue motivate the patient to comply with treatments? | 5.39 | 1.29 | To a a fairly high degree |
| 2 | 18 | Inform and educate patients and next of kin individually, taking into account time, form and content? | 5.30 | 1.61 | To a fairly high degree |
| 3 | 20 | Make sure that the patient and next of kin understand the information provided? | 5.29 | 1.43 | To a fairly high degree |
| 4 | 17 | Enable optimal participation in care and treatment, in dialogue with the patient and next of kin? | 5.22 | 1.56 | To a fairly high degree |
| 5 | 19 | Inform and educate patients and next of kin in a group, taking into account time, form and content? | 5.13 | 1.65 | To a fairly high degree |
| **Care Pedagogics %** | | | **75.23%** | **18.06** | **High** |

**Table 6. Mean, SDs, Percentage, & Interpretation of the Documentation and Administration of Nursing Care Area.**

| Rank | Item No. | Item | Mean | SD | Attitude |
|------|----------|------|------|-----|----------|
| 1 | 26 | Handle sensitive information correctly and carefully? | 5.68 | 1.30 | To a fairly high degree |
| 2 | 28 | Continuously engage in your own personal and professional competence development? | 5.55 | 1.26 | To a fairly high degree |
| 3 | 27 | Pay attention to work-related risks and actively prevent these? | 5.54 | 1.21 | To a fairly high degree |
| 4 | 24 | Carry out documentation according to current legislation? | 5.52 | 1.37 | To a fairly high degree |
| 5 | 25 | Comply with existing regulations as well as guidelines and procedures? | 5.46 | 1.24 | To a fairly high degree |
| 6 | 29 | Systematically lead, priorities delegate and coordinate nursing care within the team, based on the patient's needs and the different competencies of co-workers/staff? | 5.45 | 1.39 | To a fairly high degree |
| 7 | 22 | Make use of relevant patient records? | 5.37 | 1.23 | To a fairly high degree |
| 8 | 23 | Use information and communication technology (Horta & Victora) to support nursing care? | 5.26 | 1.44 | To a fairly high degree |
| **Documentation and Administration of Nursing Care %** | | | **78.26%** | **15.52** | **High** |

care?" has the lowest mean of 5.26±1.44 and an attitude of "to a fairly high degree." The overall percentage of documentation and administration of nursing care is 78.26%, representing a high level of competence.

For development, leadership, and organization of nursing care, Table 7 shows that item (35), "Supervise and train co-workers/staff," has the highest mean of 5.56±1.41 and an attitude of "to a fairly high degree." Item (31), "In case of a serious incident within or outside the care institution, apply emergency medical principles," has the second highest mean of 5.42±1.33 with an attitude of "to a fairly high degree." Item (30), "Act adequately in case of unprofessional conduct by staff," has the lowest mean of 5.24±1.36 with an attitude of "to a fairly high degree." The overall percentage of nursing care development and organization is 77.11%, categorized as high.

Finally, Table 8 revealed that documentation and administration of nursing care have the highest score, with 78.26%, while development, leadership, and organization of nursing care are ranked as second, with 77.11%. Care pedagogy has

**Table 7. Means, SDs, percentages, & interpretations of the development, leadership and organization of nursing care area.**

| Rank | Item No. | Item | Mean | SD | Attitude |
|------|----------|------|------|-----|----------|
| 1 | 35 | Supervise and train co-workers/staff? | 5.56 | 1.41 | To a fairly high degree |
| 2 | 31 | in case of a serious incident within or outside the care institution, apply emergency medical principles | 5.42 | 1.33 | To a fairly high degree |
| 3 | 32 | Implement new knowledge and thus promote nursing care in accordance with science and evidence-based practice? | 5.41 | 1.36 | To a fairly high degree |
| 4 | 34 | Teach, supervise and assess students? | 5.39 | 1.48 | To a fairly high degree |
| 5 | 33 | Plan, consult, inform and cooperate with other actors in the chain of care? | 5.37 | 1.33 | To a fairly high degree |
| 6 | 30 | Act adequately in case of unprofessional conduct by staff? | 5.24 | 1.36 | To a fairly high degree |
| **Development, Leadership and Organization of Nursing Care %** | | | **77.11%** | **16.83** | **High** |

**Table 8. Percentages & attitude of NPC subscale.**

| No. | Competence area | % | Attitude |
|---|---|---|---|
| 1 | Nursing care | 76.03 | **High** |
| 2 | Value-based nursing care | 76.56 | **High** |
| 3 | Medical and technical care | 76.03 | **High** |
| 4 | Care pedagogics | 75.23 | **High** |
| 5 | Documentation and administration of nursing care | 78.26 | **High** |
| 6 | Development, leadership and organization of nursing care | 77.11 | **High** |
| **NPC** | | **76.69** | **High** |

the lowest score at only 75.23%. The overall score on NPC is valued at 76.69%, indicating a high level of self-reported nursing professionalcompetence across all domains.

## Discussion

This study is the first in Jordan to examine self-reported nursing professional competence (SRNPC) among nurses working across the three main healthcare sectors who graduated from Jordanian universities. The study utilized the Nurse Professional Competence (NPC) Scale to assess nurses' competencies in six dimensions: Nursing Care, Value-based Nursing Care, Medical and Technical Care, and Care Pedagogics; Documentation and Administration of Nursing Care; and Development, Leadership, and Organization of Nursing Care.

The study's results indicated that, based on the findings of the NPC Scale, nurses felt that they were highly competent within each of the six areas. This is consistent with earlier findings from international studies (21,45). The findings of the NPC Scale were in partial agreement with the findings of Safadi et al. (46), who surveyed nursing managers and supervisors in Jordan on the competencies of nurses and reported that they rated nurses as competent in the areas studied. The discrepancies in competency ratings may be related to the different methods used in evaluating competency. The current research study utilized self-assessment, which is based on how nurses perceive their knowledge, skills, and level of confidence in their profession. Conversely, Safadi et al. (46) utilized assessments that are created by managers and are therefore more likely to be impacted by what the manager can observe about the nurse's performance as well as institutional and organizational expectations and standards. Because self-assessments are based on a nurse's assessment of their own competence, they may result in inflated competence ratings due to professional confidence, lack of external benchmarking, or the influence of social desirability. Meanwhile, manager evaluations are likely more conservative as they typically are based on observed performance rather than the nurse's perception of themselves. Because there is a difference in methodology, it is important to consider the source of the assessment when interpreting levels of competence, and hence using a multi-perspective approach, consisting of assessments from self, peers, and supervisors, may be the most effective way to measure and accurately assess the level of nursing professional competence.

[21,45,46] One factor that might explain the high levels of self-assessed competence is related to the study participants' professional backgrounds. Although about thirty-three percent of the sample demonstrated less than four years of experience, the majority of the sample had greater than four years of experience. Finally, based upon the participants' professional backgrounds, the majority of the participants had exposure to key predictors of professional competence, including taking part in structured orientation programs, working in accredited hospitals, and being actively engaged in ongoing professional development. These conditions have been proven to facilitate the development and continued retention of a nurse's professional competence.

One reason for the high level of self-reported competence in this study was the nurses' professional self-concept, as a strong professional self-concept is associated with confidence in clinical decision-making and perceived competence as

 

a nurse [47]. Therefore, nurses' perceptions of their professional identity may impact how they assess their own competence level. However, a limitation of this study is that it was based entirely on self-reported data and that nurses may inflate their level of competence because of social pressures or fears of being judged by their peers. Therefore, while self-assessment is a great way to encourage reflection and awareness, it is important to also include assessments by various individuals, such as clinical educators, supervisors, and head nurses, for a more thorough assessment of nurses' competencies. Such triangulation of data will provide a more accurate view of each nurse's overall competence. Additionally, all nurses should regularly complete multiple assessments to determine individual and organizational learning needs and deliver high-quality patient care. Additionally, evaluating nurses' competencies can also assist with professional development, as well as help organizations with strategic workforce planning and quality improvement initiatives.

Longitudinal studies should be the focus of future research to ascertain how competently someone performs and what contextual, educational, and organizational factors have an impact on development of competence. Since this study was conducted in Jordan and the Jordanian Nursing Council (JNC) is the national credentialing body that develops nursing competencies and professional standards, it is essential to develop and validate a standardized culturally appropriate competency assessment tool that is based on JNC competencies to provide the most relevant context-based measures of nursing professional competence in Jordan.

## Limitations

Limitations of the study need to be considered when reviewing the findings. First, because a cross-sectional study was used, causation cannot be established. The data represents one point in time; therefore, no indication can be given as to how the variables might change over time, nor can an inference be made about the direction of the relationship between the variables. Second, the use of convenience sampling and nurses from a small number of hospitals limits the generalizability of the findings. Thus, the sample may not be representative of all registered nurses practicing within multiple regions of Jordan and diverse healthcare environments. To increase the representativeness of future research studies, probability-based sampling methods should be employed and include greater variety in the sample selection. Thirdly, self-assessing the questionnaires creates a possibility for a respondent to answer in a biased manner. A respondent may unconsciously rate their own skill level higher than it is due to social pressure, protecting themselves in the profession, or thinking others will expect them to perform at an acceptable standard. This will create higher than actual numbers of nurses stating they are competent based on a self-assessment of their ability as opposed to actually performing at that level on a day-to-day basis. This will cause underestimating actual competency levels and therefore structure barriers (limiting the researcher or others to identify the specific areas where assisting with continuing nursing education). So be careful in interpreting the results, and for future research, develop a more comprehensive evaluation of nursing professionals based on additional methodologies such as peer review, supervisor review, and objective performance evaluation. Finally, the study only used descriptive methodology and therefore didn't provide any inferential analysis to identify or evaluate associations or predictors related to nursing competency or to generalize beyond the limited observations of trends noted in this study.

## Implications

Nursing education, practice, and research have a wide variety of potentially important implications. The results of the study demonstrate that academic institutions and clinical practice settings must work more closely together if nursing curricula are to be developed that will adequately prepare students for the increasing complexities of the modern healthcare environment. Ongoing education, structured training, and institutional support are critical aspects of a nurse's development of professional competence, as demonstrated in earlier research based on established theoretical models of clinical competence development, such as Benner's Novice to Expert Model, which describes competence as a developmental process developed through experiential learning, clinical experience and reflection on experiences gained over time [48]. Therefore, it is recommended that hospitals develop orientation and mentorship programs that include experiential

learning opportunities such as supervised practice, timely feedback, and reflective practice. Recent studies have also shown that a nurse's professional quality of life (PQL) is determined in part by a supportive work environment and organizational support, and therefore is crucial for a nurse to maintain competence, reduce burnout, and provide high quality patient care [49]. Accreditation should be a priority for all healthcare facilities due to the structured framework of accreditation that allows for ongoing quality improvement, development of a competent workforce, and continued assurance of competence. Finally, with a foundation of baseline data on nursing competence, future longitudinal/informed research on the evolution of nursing competence through the various career stages, organizational levels and health care settings will become possible

## Conclusion

This study used a descriptive design and cross-sectional survey to obtain information from nurses from multiple hospital types. The results show that nurses, in general, have high levels of competence with respect to both nursing care and value-based nursing and medical-technical nursing care. Therefore, according to these data, both nursing educational programs in Jordan should strengthen their curriculums in care pedagogies, psychosocial nursing, group patient education, and culturally competent practice through nursing educational programs, while healthcare facilities should provide ongoing professional development for each nurse to enhance their skills regarding patient education, communication, and holistic care delivery. The study contributes to the ongoing discussions on professional standards and patient safety within the Jordanian healthcare system.

## Supporting information

**S1 File. Excel file with de-identified study data used in the analyses.**
(XLSX)

## Author contributions

**Conceptualization:** Alaadin Alharaizahe, Intima Alrimawi, Hekmat Yousef Al-Akash, Abedalmajeed Shajrawi, Abdul-Monim Batiha, Osama Al-Kouri, Manar Abu-Abbas, Haitham Khatatbeh, Ahmad Rajeh Saifan.

**Data curation:** Alaadin Alharaizahe, Intima Alrimawi, Hekmat Yousef Al-Akash, Abedalmajeed Shajrawi, Abdul-Monim Batiha, Osama Al-Kouri, Manar Abu-Abbas, Haitham Khatatbeh, Ahmad Rajeh Saifan.

**Formal analysis:** Alaadin Alharaizahe, Intima Alrimawi, Hekmat Yousef Al-Akash, Abedalmajeed Shajrawi, Abdul-Monim Batiha, Osama Al-Kouri, Manar Abu-Abbas, Haitham Khatatbeh, Ahmad Rajeh Saifan.

**Investigation:** Alaadin Alharaizahe, Intima Alrimawi, Abdul-Monim Batiha, Osama Al-Kouri, Manar Abu-Abbas, Haitham Khatatbeh, Ahmad Rajeh Saifan.

**Methodology:** Alaadin Alharaizahe, Intima Alrimawi, Hekmat Yousef Al-Akash, Abedalmajeed Shajrawi, Abdul-Monim Batiha, Osama Al-Kouri, Manar Abu-Abbas, Haitham Khatatbeh, Ahmad Rajeh Saifan.

**Resources:** Abdul-Monim Batiha, Haitham Khatatbeh, Ahmad Rajeh Saifan.

**Software:** Alaadin Alharaizahe, Intima Alrimawi, Hekmat Yousef Al-Akash, Abedalmajeed Shajrawi, Abdul-Monim Batiha, Osama Al-Kouri, Manar Abu-Abbas, Haitham Khatatbeh, Ahmad Rajeh Saifan.

**Supervision:** Ahmad Rajeh Saifan.

**Validation:** Alaadin Alharaizahe, Intima Alrimawi, Hekmat Yousef Al-Akash, Abedalmajeed Shajrawi, Abdul-Monim Batiha, Osama Al-Kouri, Manar Abu-Abbas, Haitham Khatatbeh, Ahmad Rajeh Saifan.

**Visualization:** Alaadin Alharaizahe, Intima Alrimawi, Hekmat Yousef Al-Akash, Abedalmajeed Shajrawi, Abdul-Monim Batiha, Osama Al-Kouri, Manar Abu-Abbas, Haitham Khatatbeh, Ahmad Rajeh Saifan.

**Writing – original draft:** Alaadin Alharaizahe, Intima Alrimawi, Hekmat Yousef Al-Akash, Abedalmajeed Shajrawi, Abdul-Monim Batiha, Osama Al-Kouri, Manar Abu-Abbas, Haitham Khatatbeh, Ahmad Rajeh Saifan.

**Writing – review & editing:** Alaadin Alharaizahe, Intima Alrimawi, Hekmat Yousef Al-Akash, Abedalmajeed Shajrawi, Abdul-Monim Batiha, Osama Al-Kouri, Manar Abu-Abbas, Haitham Khatatbeh, Ahmad Rajeh Saifan.

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
