## [Decision Letter · Decision Letter 0]

6 Nov 2025

Dear Dr. Alrimawi,

Thank you for submitting your manuscript to PLOS ONE. After careful consideration, we feel that it has merit but does not fully meet PLOS ONE’s publication criteria as it currently stands. Therefore, we invite you to submit a revised version of the manuscript that addresses the points raised during the review process.

We look forward to receiving your revised manuscript.

Kind regards,

Ahmad H. Al-Nawafleh, Ph.D, MPA, CI, RN

Academic Editor

PLOS ONE

Journal Requirements:

1. Please ensure that your manuscript meets PLOS ONE's style requirements, including those for file naming. The PLOS ONE style templates can be found at.

https://journals.plos.org/plosone/s/file?id=wjVg/PLOSOne_formatting_sample_main_body.pdf and.

2. Please include a complete copy of PLOS’ questionnaire on inclusivity in global research in your revised manuscript. Our policy for research in this area aims to improve transparency in the reporting of research performed outside of researchers’ own country or community. The policy applies to researchers who have travelled to a different country to conduct research, research with Indigenous populations or their lands, and research on cultural artefacts. The questionnaire can also be requested at the journal’s discretion for any other submissions, even if these conditions are not met.

Please find more information on the policy and a link to download a blank copy of the questionnaire here: https://journals.plos.org/plosone/s/best-practices-in-research-reporting.

Please upload a completed version of your questionnaire as Supporting Information when you resubmit your manuscript.

https://www.sciedupress.com/journal/index.php/jnep/article/viewFile/22609/14362

https://search.informit.org/doi/abs/10.3316/informit.223994226712264

In your revision ensure you cite all your sources (including your own works), and quote or rephrase any duplicated text outside the methods section. Further consideration is dependent on these concerns being addressed.

5. Thank you for stating the following financial disclosure:.

“Funding support is in part from Georgetown University Medical Center.“

6. Thank you for stating in your Funding Statement:.

“Funding support is in part from Georgetown University Medical Center.“

7. Please note that funding information should not appear in the Acknowledgments section or other areas of your manuscript. We will only publish funding information present in the Funding Statement section of the online submission form. Please remove any funding-related text from the manuscript.

8. We note that your Data Availability Statement is currently as follows:.

“All relevant data are within the manuscript and its Supporting Information files.”

9. Your ethics statement should only appear in the Methods section of your manuscript. If your ethics statement is written in any section besides the Methods, please delete it from any other section.

10. We note you have included a table to which you do not refer in the text of your manuscript. Please ensure that you refer to Table 1 in your text; if accepted, production will need this reference to link the reader to the Table.

**Additional Editor Comments:**

Dear Authors,

Your study received responses from the reviewers and require some attention to enhance its rigor. Please check the reviewers comments and responde to each comment according to the journal instructions.

Looking forward to see your responses

All the best

Reviewers' comments:

Reviewer's Responses to Questions

**Comments to the Author**

1. Is the manuscript technically sound, and do the data support the conclusions?

Reviewer #1: Yes

Reviewer #2: Yes

Reviewer #3: Yes

Reviewer #4: Yes

2. Has the statistical analysis been performed appropriately and rigorously?

Reviewer #1: Yes

Reviewer #2: Yes

Reviewer #3: N/A

Reviewer #4: Yes

3. Have the authors made all data underlying the findings in their manuscript fully available?

Reviewer #1: Yes

Reviewer #2: Yes

Reviewer #3: Yes

Reviewer #4: Yes

4. Is the manuscript presented in an intelligible fashion and written in standard English?

Reviewer #1: Yes

Reviewer #2: Yes

Reviewer #3: Yes

Reviewer #4: Yes

Reviewer #1: This study provides valuable insight into the self-perceived competence of registered nurses in Jordan, addressing an important gap in the national nursing literature. the paper contributes meaningfully to efforts aimed at enhancing nursing quality and patient safety in Jordan.

Reviewer #2: This manuscript presents a valuable and timely investigation into self-reported nursing competence in Jordan. The study addresses a clear gap in the literature, utilizes a validated instrument, and has a robust sample size. The findings are relevant for nursing education and practice in the region. However, several issues require attention to enhance the manuscript's scientific rigor, clarity, and impact before it is suitable for publication.

Reviewer #3: Abstract Clarity: While the abstract provides a good overview, consider rephrasing the sentence "Despite nurses' essential contributions, limited research has examined self-reported nursing competence (SRNC) in Jordan" to emphasize the gap in research more directly. For example, "Despite the critical role of nurses, research on self-reported nursing competence (SRNC) in Jordan remains limited."

Keyword Refinement: Consider adding "Cross-Sectional Study" to the keywords to improve searchability.

Ethical Statement: Ensure the ethics statement includes a mention of informed consent and that participants were aware of their right to withdraw from the study at any time.

Sample Description: Clarify the phrase, "years of experience ranged from 0 to 4 years." Does this mean the range was very limited, or should this be a higher number? Furthermore, specify whether the experience is in nursing.

Statistical Analysis: Briefly mention any specific statistical tests used beyond descriptive statistics (e.g., t-tests, ANOVA) if applicable.

Table Formatting: Ensure all tables are clearly labelled and that column headings are concise and easily understood. Check for consistency in decimal places.

Discussion – Comparison to Other Studies: When comparing findings to Safadi et al. (43), elaborate further on the differences in perspectives (self-assessment vs. manager assessment) and how these distinctions might explain any discrepancies.

Limitations – Response Bias: Elaborate on the possible consequences of response bias in the self-assessment questionnaires. How might this specifically affect the interpretation of the results?

Typos and Grammar: Proofread carefully for any remaining typos or grammatical errors.

Reviewer #4: Specific Comments and Suggestions

Introduction–Jordan Healthcare System Context:

Specific Comment: The introduction would benefit from a brief overview of the Jordanian healthcare system. Consider adding a sentence or two about its structure (e.g., public vs. private provision), key challenges (e.g., resource constraints, access disparities), and any recent reforms or initiatives. This will help readers understand the context in which nursing competence is being assessed.

Suggestion: Consult sources like the World Health Organization (WHO) or recent publications on healthcare in Jordan to gather this contextual information.

Research Question Refinement:

Specific Comment: The current research question is broad. Rephrasing it to be more specific will improve the focus of the study.

Suggestion: Instead of "What is the level of self-reported professional nursing competence among registered nurses in Jordan?" try: "What is the self-reported level of professional nursing competence among registered nurses in Jordan across different domains of practice, as measured by the Nursing Professional Competence (NPC) scale?" This clarifies the measurement instrument and the scope of the assessment.

Instrumentation - NPC Scale Description:

Specific Comment: The Methods section lacks detail about the domains assessed by the NPC scale. Readers need to know what specific aspects of nursing competence were evaluated.

Suggestion: Add a sentence or two describing the six competence areas measured by the NPC scale (Nursing Care, value-based care, etc.). For example: "The NPC scale assesses six key areas of nursing competence: Nursing Care, Value-Based Care, Medical and Technical Care, Care Pedagogics, Documentation and Administration, and Development, Leadership, and Organization of Care."

Results - Examples of High and Low-Scoring Items:

Specific Comment: It's difficult to interpret the meaning of the high and low overall scores without knowing which specific items contributed most to those scores.

Suggestion: In the Results section, when discussing the high- and low-scoring areas, provide a few examples of the specific items from the NPC scale that received the highest and lowest ratings. For example: "The highest-rated item was 'Document the patient's physical condition', while the lowest-rated item was 'Address the patient's psychological condition.'"

Conclusion – Practical Implications:

Specific Comment: The conclusion should offer more concrete recommendations based on the study's findings. What specific actions can be taken to improve nursing competence in Jordan?

Suggestion: Expand the conclusion to include actionable recommendations for nursing education and practice. For example: "Based on these findings, nursing education programs in Jordan should consider strengthening curricula in areas such as [specific area needing improvement], while healthcare institutions should focus on providing ongoing professional development opportunities in [specific area needing improvement]."

References – Consistent Formatting:

Specific Comment: The reference list may have inconsistencies in formatting.

Suggestion: Carefully review all references to ensure they adhere to the PLOS ONE style guidelines. Pay attention to journal abbreviations, capitalization, italics, and the order of elements.

Limitations - Generalizability:

Specific Comment: The study's generalizability may be limited by the sampling method.

Note in the Limitations section that the findings may not represent all Jordanian nurses due to the convenience sampling method and the specific hospitals included in the study. For example: "The use of convenience sampling limits the generalizability of the findings to all nurses in Jordan. Future research should employ random sampling methods to obtain a more representative sample."

**Do you want your identity to be public for this peer review?** For information about this choice, including consent withdrawal, please see our Privacy Policy

Reviewer #1: No

Reviewer #2: No

Reviewer #3: No

Reviewer #4: No

---

## [Author Response · Author response to Decision Letter 1]

5 Jan 2026

See the attached 'Response to Reviewers' document

---

## [Editor Report · Decision Letter 1]

12 Jan 2026

Self-Reported Nursing Competence Among Registered Nurses in Jordan: A Cross-Sectional Study

PONE-D-25-57366R1

Dear Dr. Alrimawi,

We’re pleased to inform you that your manuscript has been judged scientifically suitable for publication and will be formally accepted for publication once it meets all outstanding technical requirements.

Kind regards,

Ahmad H. Al-Nawafleh, Ph.D, MPA, CI, RN

Academic Editor

PLOS One

Additional Editor Comments (optional):

Dear Authors,

Thank you for your amendments. I believe the work is satisfactory. Except, there are indications of using AI in your amendments. I prefer, that you work on them only and resubmit.

Wish you all the best

Reviewers' comments:

<!--a=1<!--a=1

---

## [Editor Report · Acceptance letter]

25 Nov 2025

PONE-D-25-57366R1

PLOS One

Dear Dr. Alrimawi,

I'm pleased to inform you that your manuscript has been deemed suitable for publication in PLOS One. Congratulations! Your manuscript is now being handed over to our production team.

Kind regards,

on behalf of

Prof. Ahmad H. Al-Nawafleh

Academic Editor

PLOS One